# Two Years of Genomic Surveillance in Belgium during the SARS-CoV-2 Pandemic to Attain Country-Wide Coverage and Monitor the Introduction and Spread of Emerging Variants

**DOI:** 10.3390/v14102301

**Published:** 2022-10-20

**Authors:** Lize Cuypers, Simon Dellicour, Samuel L. Hong, Barney I. Potter, Bruno Verhasselt, Nick Vereecke, Laurens Lambrechts, Keith Durkin, Vincent Bours, Sofieke Klamer, Guillaume Bayon-Vicente, Carl Vael, Kevin K. Ariën, Ricardo De Mendonca, Oriane Soetens, Charlotte Michel, Bertrand Bearzatto, Reinout Naesens, Jeremie Gras, Anne Vankeerberghen, Veerle Matheeussen, Geert Martens, Dagmar Obbels, Ann Lemmens, Bea Van den Poel, Ellen Van Even, Klara De Rauw, Luc Waumans, Marijke Reynders, Jonathan Degosserie, Piet Maes, Emmanuel André, Guy Baele

**Affiliations:** 1National Reference Centre for Respiratory Pathogens, Department of Laboratory Medicine, University Hospitals Leuven, 3000 Leuven, Belgium; 2Spatial Epidemiology Lab (SpELL), Université Libre de Bruxelles, 1000 Brussels, Belgium; 3Department of Microbiology, Immunology and Transplantation, Rega Institute, KU Leuven, 3000 Leuven, Belgium; 4Department of Diagnostic Sciences, Ghent University Hospital, Ghent University, 9000 Ghent, Belgium; 5PathoSense BV, Faculty of Veterinary Medicine, Ghent University, 9820 Merelbeke, Belgium; 6HIV Cure Research Center, Department of Internal Medicine and Pediatrics, Ghent University Hospital, Ghent University, 9000 Ghent, Belgium; 7BioBix, Department of Data Analysis and Mathematical Modelling, Faculty of Bioscience Engineering, Ghent University, 9000 Ghent, Belgium; 8Laboratory of Human Genetics, GIGA Research Institute, 4000 Liège, Belgium; 9Department of Human Genetics, University Hospital of Liège, 4000 Liège, Belgium; 10Scientific Directorate of Epidemiology and Public Health, Sciensano, 1050 Brussels, Belgium; 11Department of Proteomic and Microbiology, Research Institute for Biosciences, University of Mons, 7000 Mons, Belgium; 12Clinical Laboratory, AZ Klina, 2930 Brasschaat, Belgium; 13Virology Unit, Department of Biomedical Sciences, Institute of Tropical Medicine Antwerp, 2000 Antwerp, Belgium; 14Faculty of Pharmaceutical, Biomedical and Veterinary Sciences, Department of Biomedical Sciences, University of Antwerp, 2000 Antwerp, Belgium; 15Department of Microbiology, CUB-Hôpital Erasme, Université Libre de Bruxelles, 1000 Brussels, Belgium; 16Department of Microbiology and Infection Control, Universitair Ziekenhuis Brussel, Vrije Universiteit Brussel, 1090 Brussels, Belgium; 17Department of Microbiology, Laboratoire Hospitalier Universitaire de Bruxelles (LHUB-ULB), 1000 Brussels, Belgium; 18Center for Applied Molecular Technologies (CTMA), Institut de Recherche Expérimentale et Clinique (IREC), Université Catholique de Louvain (UCLouvain), 1000 Brussels, Belgium; 19Department of Medical Microbiology, Ziekenhuis Netwerk Antwerpen, 2020 Antwerp, Belgium; 20Institute of Pathology and Genetics (IPG), 6041 Gosselies, Belgium; 21Laboratory of Molecular Biology, Campus Aalst-Asse-Ninove, Onze-Lieve-Vrouwziekenhuis, 9300 Aalst, Belgium; 22Laboratory of Medical Microbiology, Department of Microbiology, Vaccine & Infectious Disease Institute (VAXINFECTIO), University of Antwerp, 2610 Wilrijk, Belgium; 23Department of Laboratory Medicine, AZ Delta General Hospital, 8800 Roeselare, Belgium; 24Clinical Laboratory, Imelda Hospital, 2820 Bonheiden, Belgium; 25Laboratory of Clinical Biology, AZ Sint-Maarten Hospital, 2800 Mechelen, Belgium; 26Clinical Laboratory, General Hospital Jan Portaels, 1800 Vilvoorde, Belgium; 27Clinical Laboratory of Microbiology, HH Hospital Lier, 2500 Lier, Belgium; 28Laboratory of Clinical Biology, AZ Sint Lucas Hospital, 9000 Ghent, Belgium; 29Clinical Laboratory, Jessa Hospital, 3500 Hasselt, Belgium; 30Department of Laboratory Medicine, Medical Microbiology, AZ Sint-Jan Bruges-Ostend AV, 8000 Bruges, Belgium; 31Federal Testing Platform COVID-19, Department of Laboratory Medicine, CHU UCL Namur, 5530 Yvoir, Belgium; 32Next Generation Sequencing Platform, Molecular Diagnostic Center, CHU UCL Namur, 5530 Yvoir, Belgium; 33Federal Testing Platform COVID-19, Department of Laboratory Medicine, University Hospitals Leuven, 3000 Leuven, Belgium; 34Laboratory of Clinical Microbiology, Department of Microbiology, Immunology and Transplantation, KU Leuven, 3000 Leuven, Belgium

**Keywords:** SARS-CoV-2, COVID-19, Belgium, genomic surveillance, next-generation sequencing, variants of concern

## Abstract

An adequate SARS-CoV-2 genomic surveillance strategy has proven to be essential for countries to obtain a thorough understanding of the variants and lineages being imported and successfully established within their borders. During 2020, genomic surveillance in Belgium was not structurally implemented but performed by individual research laboratories that had to acquire the necessary funds themselves to perform this important task. At the start of 2021, a nationwide genomic surveillance consortium was established in Belgium to markedly increase the country’s genomic sequencing efforts (both in terms of intensity and representativeness), to perform quality control among participating laboratories, and to enable coordination and collaboration of research projects and publications. We here discuss the genomic surveillance efforts in Belgium before and after the establishment of its genomic sequencing consortium, provide an overview of the specifics of the consortium, and explore more details regarding the scientific studies that have been published as a result of the increased number of Belgian SARS-CoV-2 genomes that have become available.

## 1. Introduction

The continued accumulation of SARS-CoV-2 infections across the world keeps posing significant threats to public health. Since the start of the pandemic, many countries sought to avoid or control a next wave of infections using non-pharmaceutical interventions (including social distancing, mask wearing and partial or complete lockdowns) to avoid flooding hospitals with patients and to keep medical care facilities from collapsing. The key motivation behind many of these measures was to guarantee high-quality health care and limit delays to essential treatments (e.g., surgery, chemotherapy and/or radiation therapy) for all patients. After an initial wave of infections in 2020, countries sought to find a balance between the impact of the imposed preventative measures on economic and social activities, and the often severe and potentially long-term impact of COVID-19 on public health (e.g., long COVID), and economic and social activities [1]. Since the end of 2020, vaccination campaigns have been deployed around the world, and an increasing number of vaccines continues to be approved by public health authorities, while ongoing research efforts monitor the evolution and spread of SARS-CoV-2 at both national and international levels.

In the midst of combating the pandemic, the repeated emergence of new mutations of the SARS-CoV-2 genome in different countries has been and still is a cause for concern. Accumulation of mutations is a natural consequence of the abundant replication of the virus worldwide, and it is to be expected that a subset of these mutations is progressively selected, leading to the emergence of new variants. Such emerging variants might harbor advantages with regard to transmissibility, immune escape and/or other fitness features in comparison to other circulating strains, and they are therefore to be carefully monitored and evaluated. The D614G mutation on the spike protein was the first mutation suspected to lead to a higher transmissibility of SARS-CoV-2. Rarely occurring before March 2020, this mutation became increasingly common as the pandemic spread, occurring in over 74% of all published sequences by June 2020 [2,3]. Although this variant nearly completely replaced the original Wuhan variant, it is still debated as to whether this mutation did indeed lead to increased transmissibility, with certain studies having suggested only a moderate impact on transmissibility [4,5].

In late 2020, the UK faced a rapid increase in COVID-19 cases in South East England, leading to enhanced epidemiological and virological investigations [6], which revealed a novel SARS-CoV-2 variant, now referred to as Alpha (lineage B.1.1.7). This variant was defined by a set of 23 mutations, 14 amino acid changes and three deletions, compared to the original Wuhan isolate. Most notably, the N501Y mutation and the HV 69–70 deletion are present in lineage B.1.1.7 [7], which had been growing in frequency since November 2020. Preliminary analyses in the UK rapidly suggested that this variant was significantly more transmissible than previously circulating variants, with an estimated potential to increase the effective reproductive number (Re) by a value ranging between 0.4 and 0.7 [8].

The SARS-CoV-2 variant of concern (VOC) Alpha (lineage B.1.1.7) was identified using a broad high-throughput sequencing (HTS) strategy implemented in the UK, where up to 10% of all SARS-CoV-2 PCR-positive samples were being sequenced at the time. This strategy also enabled the UK to detect two cases of the Beta VOC (lineage B.1.351), which was first detected in South Africa. Impacting mostly southern African countries and defined by eight mutations in the spike protein, including three substitutions (K417N, E484K and N501Y) at residues in its receptor-binding domain that may have functional importance, Beta showed rapid expansion and displacement of other lineages in several regions [9]. However, the impact of this VOC in Belgium remained rather limited compared to the other VOCs.

First detected in Brazil, and quickly following in the footsteps of Beta (lineage B.1.351), the Gamma VOC (lineage P.1) was characterized by 17 mutations, including a trio of mutations in the spike protein (K417T, E484K, and N501Y) associated with increased binding to the human ACE2 (angiotensin-converting enzyme 2) receptor. Faria et al. [10] estimated that P.1 may have been 1.7 to 2.4 times more transmissible than local non-P.1 lineages, and that previous (non-P.1) infection provides 54 to 79% of the protection against infection with P.1 than it provides against non-P.1 lineages. The Gamma VOC mostly affected southern American countries, but, like Beta, had only limited impact on the pandemic situation in Belgium.

Starting its rise to become the global dominant lineage in April 2021, the Delta VOC (lineage B.1.617.2) was first detected in India and bears the L452R spike receptor-binding motif (RBM) substitution, previously reported to confer increased infectivity and a modest loss of susceptibility to neutralizing antibodies. Mlcochova et al. [11] found that increased replication fitness and reduced sensitivity of SARS-CoV-2 B.1.617.2 to neutralizing antibodies contributed to the rapid increase in B.1.617.2 cases, compared to B.1.1.7 and other lineages. The authors also demonstrated the evasion of neutralizing antibodies by a B.1.617.2 live virus with sera from convalescent patients, as well as sera from individuals vaccinated with two different vaccines, and reported vaccine breakthrough infections in healthcare workers in three hospitals, demonstrating reduced vaccine effectiveness against B.1.617.2.

Since mid-January 2022, the Omicron VOC (lineage B.1.1.529/BA.1) has become the dominant lineage in most countries worldwide, with a growing tendency to displace lineage B.1.617.2. Having been detected in November 2021 by genomic surveillance teams in South Africa and Botswana, the Omicron VOC carries over 30 mutations in the spike glycoprotein, which Martin et al. [12] predicted would influence antibody neutralization and spike function. Viana et al. [13] were the first to describe the genomic profile and early transmission dynamics of Omicron, showing rapid spread of the Omicron VOC in regions with high levels of population immunity.

The continued emergence of novel SARS-CoV-2 lineages and VOCs is testament to the importance of performing adequate genomic surveillance in countries around the world. Toward the end of 2020, the ECDC emphasized the reinforcement of HTS to facilitate the detection of these and (re-)emerging variants [14]. Sample collection should aim for a robust representation of the population (geographic distribution, age groups, etc.) to create a representative baseline surveillance, but could also have a focus on areas or populations associated with a rapid increase in incidence or reports of increased severity. The ECDC and the WHO further insisted that member states deposit their sequences to the international database GISAID [15] to make them available to the entire scientific community [14]. Additionally, associated metadata should be shared and annotated in a consistent manner to facilitate analyses and accompanying visualization. Already, in 2020, the ECDC offered their member states a case-based reporting system for the reporting of genomic SARS-CoV-2 results, with a focus on the emerging VOCs, coupled with epidemiologic COVID-19 case-based reporting through national public health institutes.

As a result, and starting in 2021, countries around the world intensified their efforts toward genomic surveillance in order to detect and monitor the co-circulating VOCs Alpha, Beta, Gamma [16], Delta and Omicron in their population, recognizing the need for ongoing and continuous sequencing during the SARS-CoV-2 pandemic [17]. We here describe the genomic surveillance efforts in Belgium, both before and after the establishment of a nationwide genomic surveillance consortium. We first discuss project-based research efforts from before systematic funding for whole-genome sequencing (WGS) became available, followed by the structure and workflow of the Belgian genomic surveillance consortium. These efforts opened up many opportunities for joint projects and research studies, which we discuss in the respective sections below.

## 2. Prior to the National Genomic Surveillance Initiative

In 2020, SARS-CoV-2 sequencing efforts in Belgium were still largely research-oriented, as no dedicated funding (from the federal government) had been reserved for genomic surveillance efforts. Within the Flemish Region, two general COVID-19 funding calls were launched during 2020, totaling EUR 2.5 million each, to be distributed among ten research projects. However, the eligible research projects were a mix of social, economic, and public health applications, all competing with vaccine research and genomic sequencing projects. Within the Wallonia-Brussels Federation, one call for Exceptional Research Projects (PER) Coronavirus and another for Urgent Research Credits (CUR) Coronavirus were both launched in 2020, but no dedicated funding was set aside for genomic sequencing here either. As a result, only two SARS-CoV-2 sequencing projects were granted within Belgium (one in the Flemish Region, and one in the Wallonia-Brussels Federation) during 2020, which resulted in different coexisting sampling strategies: UZ/KU Leuven (acting as part of the National Reference Center—NRC—for Respiratory Pathogens) collected samples from all over the country, while the University of Liège focused on samples originating from the province of Liège, the University of Ghent on samples from the provinces of East and West Flanders, and the Institute of Tropical Medicine Antwerp on samples from the province of Antwerp. This biased sampling strategy is apparent when visualizing a large number of the publicly available Belgian sequences in GISAID (see Figure 1). As a result, Figure 1 shows clear geographical gaps in Belgium’s spatial sequencing coverage during the first year of the pandemic, on top of a lack of temporal continuity, both being the consequences of project-based funding and a non-centralized approach to SARS-CoV-2 genomic surveillance efforts.

As part of the NRC for Respiratory Pathogens, UZ/KU Leuven was the first laboratory in Belgium to diagnose COVID-19 cases by PCR, identifying the first Belgian case on 3 February 2020. From the start and throughout the first year of the pandemic, a geographically heterogeneous share of the samples collected on a national level (Figure 1) was analyzed in the clinical laboratory of UZ Leuven, providing a unique opportunity for a national genomic surveillance initiative. Although without structural funding at that time, over 750 complete SARS-CoV-2 genomes of samples originating from the first epidemiological wave (March to June 2020) in Belgium were sequenced to characterize the temporal and geographic distribution of the COVID-19 pandemic in Belgium through phylogenetic and variant analysis. Wawina-Bokalanga et al. [18] showed the presence of the major SARS-CoV-2 clades (G, GH and GR) and lineages circulating in Belgium at that time. The continuation of this initiative to sequence a share of the positive samples, conducted by the different sequencing centers at that time, resulted in a total of over 3700 genomes in Belgium by the end of 2020.

In order to inform authorities, the scientific community, and the general public on the evolution and spread of SARS-CoV-2 in Belgium, we used the available genomes to construct our first Belgian Nextstrain instance, which became publicly available on 8 January 2021 (see Figure 2). Nextstrain’s joint temporal and spatial visualizations integrate sequence data with geographic information, lineage nomenclature, and mutations of interest to show how the pandemic unfolded over time, delivering important insights to health professionals, epidemiologists, virologists, and the general public via easily shareable links (for example, through social media) [19].

Further, the availability of such a large collection of genomes resulted in various SARS-CoV-2 studies to which members of the genomic surveillance network were able to contribute, before a national initiative was even set up. These studies primarily focused on the evolution and spread of SARS-CoV-2 in Belgium and Europe [20]. Dellicour et al. [21] developed a phylodynamic workflow that combines maximum-likelihood phylogenetic inference with Bayesian phylogeographic inference to rapidly analyze the spatiotemporal dispersal history and dynamics of SARS-CoV-2 lineages. At the time of their analysis (10 June 2020), Belgium had one of the highest spatial densities of available SARS-CoV-2 genomes—with 740 genomes sequenced, owing to the aforementioned research initiatives—which allowed Dellicour et al. [21] to apply their method on the pandemic situation in Belgium and identify a large number of lineage introductions into the country. Making use of this workflow and focusing on the second European pandemic wave (March–November 2020), Bollen et al. [22] further employed Bayesian phylogeographic inference on each clade occurring in the province of Liège. The authors focused on inferring the regional dispersal history of viral lineages associated with three specific mutations on the spike protein (S98F, A222V and S477N) and quantifying their relative importance through time.

Several studies focused on very specific settings to study SARS-CoV-2 infections among Belgian residents. At the beginning of May 2020, 22 out of 70 Belgian soldiers deployed to a military education and training center in Maradi, Niger, developed mild COVID-19 compatible symptoms [23]. Immediately upon their return to Belgium, and two weeks later, all seventy soldiers were tested for SARS-CoV-2 RNA and antibodies. Nine soldiers had at least one positive COVID-19 diagnostic test result. Five of them exhibited COVID-19 symptoms (mainly anosmia, ageusia, and fever), while four were asymptomatic. Conventional and genomic epidemiological data suggested that these infections had a most recent common ancestor with African origin and that the Belgian military service men were infected through contact with locals.

Using postmortem COVID-19 cases to perform detailed virological analysis could provide proof of viremia and presence of replication-competent SARS-CoV-2 in extrapulmonary organs of immunocompromised patients, including heart, kidney, liver and spleen [24]. In parallel, organ-specific SARS-CoV-2 genomic diversity and mutations of concern have been identified prior to the emergence of VOCs. Based on disease duration and viral loads in plasma and lungs, two stages of fatal disease evolution were addressed by Van Cleemput et al., providing insights about the pathogenesis and intra-host evolution of SARS-CoV-2 in immunocompromised patients.

Nursing homes constitute a highly vulnerable setting for the introduction and spread of SARS-CoV-2 among their inhabitants, staff, and visitors. Vuylsteke et al. [25] describes a massive outbreak of COVID-19 after a cultural event in a nursing home in Flanders, Belgium, at the end of 2020. Within days of the event, nursing home residents started to display symptoms, and the outbreak spread rapidly within the nursing home, leading to a total of 127 residents and 40 staff members being diagnosed with SARS-CoV-2 since the beginning of the outbreak. Vuylsteke et al. [25] claim that airborne transmission was the most plausible explanation for the massive intra-facility spread, which underscores the importance of ventilation and air quality for the prevention of future outbreaks in such closed facilities.

Some of the earliest SARS-CoV-2 genomes sequenced in Belgium at the start of the pandemic were used to develop a novel Bayesian phylogeographic inference methodology that is able to exploit the individual travel histories of infected patients. Lemey et al. [26] show that making use of such travel history data yields more realistic hypotheses of virus spread, and can suggest alternative routes of virus migration that are plausible within the epidemiological context but are not apparent when sampling efforts are limited or highly heterogeneous in the affected countries.

Following this study, Lemey et al. [27] built a phylogeographical model to evaluate how newly introduced lineages, as opposed to the rekindling of persistent lineages, contributed to the resurgence of COVID-19 in Europe in late summer, 2020. The authors informed their model using genomic, mobility, and epidemiological data from 10 European countries and estimated that in many countries, more than half of the lineages circulating in the late summer of 2020 resulted from new introductions, and that the success in onward transmission of newly introduced lineages was negatively associated with the local incidence of COVID-19 during this period.

## 3. Setting Up the National Genomic Surveillance Platform for SARS-CoV-2 in Belgium

In late 2020, as a response to the need to implement a more efficient genome surveillance program in Belgium, due to increasing concerns about the Alpha, Beta and Gamma VOCs, a coordinated sequencing and surveillance strategy was set up by the NRC UZ/KU Leuven in collaboration with the national public health institute, Sciensano. Rapidly, a large national sequencing consortium combined forces to substantially scale up the national sequencing capacity in the early months of 2021. This coordinated strategy focused on nationwide genomic sequencing in order to detect transmission hotspots and the possible (re-)emergence of recent variants first detected in the UK and South Africa, or any other variants for that matter. Enhanced molecular surveillance was aimed at monitoring SARS-CoV-2 diversity more closely in space and time across the country. Specifically, this initiative enabled the tracking of VOC frequency, as well as their circulation within the country. Furthermore, it provided an early warning system in the case of the emergence of potentially more transmissible, virulent, or vaccine-escaping strains. We note that while the official start of the nationwide genomic surveillance initiative was at the end of February 2021, many labs had already started their sequencing efforts at the beginning of the year, thereby strengthening surveillance in anticipation of the nationwide initiative. This can be seen in Figure 1, which shows a strong increase in available genome sequences across all Belgian provinces since the start of the national SARS-CoV-2 genomic surveillance initiative.

At the inter-ministerial conference for public health on 20 January 2021, it was decided that special attention should be given to the proactive detection and monitoring of circulating variants of the SARS-CoV-2 virus. A conceptual note was presented at the insurance committee on 22 February 2021, followed by the drafting of a detailed two-pillar model of surveillance efforts: a national genomic surveillance consortium, fueled by a sentinel laboratory network for the supply of a representative share of their positive samples. In mid-March, a call was launched to all laboratories in Belgium to apply to be part of the national genomic surveillance consortium for SARS-CoV-2, coordinated by the NRC UZ/KU Leuven with the support of Sciensano. Following this candidacy call, a network of 17 laboratory network partners was initiated to be responsible for the SARS-CoV-2 genomic surveillance in Belgium, of which the majority of centers signed a convention for reimbursement purposes (Figure 3). To this end, based on a predefined required sequencing capacity per Belgian province (based on the number of inhabitants rather than infections, as the latter would fluctuate over time), a minimum of 1150 genome sequences needed to be generated per week, roughly aiming to sequence at least 5% of all positive cases diagnosed within Belgium.

The year following the start of the nationwide genomic surveillance initiative in Belgium saw an important increase in the sequencing coverage of positive cases, rising from 1% to nearly 4% (Figure 4). While the initial sequencing coverage of positive cases of 1% already put Belgium at a recently proposed threshold/benchmark for rapidly detecting circulating SARS-CoV-2 variants through random sampling, this increase put Belgium closer towards ensuring the rapid detection of viral lineages, according to a recent simulation study [28]. This increase puts Belgium just below several other European countries in terms of coverage of positive cases, including Germany, Austria, Norway, and Finland, but ahead of the Netherlands, France, Portugal, and Spain. Of note, this sequencing coverage of positive cases has been highly variable throughout the pandemic, directly related to the number of cases associated with the different pandemic waves of infections. During times when there has been a relatively low number of infections, sequencing coverage of positive cases was shown to reach nearly 25% of cases (Figure 5 and Figure 6), whereas during epidemic peaks of infections, the imposed threshold of 5% could not be reached consistently.

In Europe, Iceland, the UK, and Denmark have been particularly credited for their massive genomic sequencing efforts throughout the pandemic (Figure 4), with Denmark deciding to sequence every positive case during certain periods throughout the pandemic, and Iceland sequencing up to 90% of positive cases during the first year of the pandemic. In the UK, the COVID-19 Genomics UK Consortium (COG-UK), established in April 2020, undertook sequencing of SARS-CoV-2 samples from a target of 10% of confirmed positive cases, which translated to an average of 6800 sequences per week during its first year (www.cogconsortium.uk (accessed on 22 June 2021)). For comparison, during the same time period, Denmark averaged approximately 1000 sequences per week and the Netherlands averaged 370 sequences per week. Prior to the establishment of the Belgian consortium, in terms of sequencing efforts within Europe, Iceland and Denmark attained higher numbers than the UK, with a coverage of 71% and 23% of confirmed positive cases, respectively. Meanwhile, Luxemburg sequenced approximately 6%, Finland, Norway and Switzerland roughly 2%, and Ireland, Belgium and the Netherlands nearly 1% (Figure 4).

The two-pillar model of genomic surveillance consists of, on the one hand, an unbiased and random sequencing of cases, called baseline (or passive) surveillance, while on the other hand, the pillar of active surveillance focused on specific questions, populations, or settings, as determined through official sequencing indications proposed by the risk-assessment group (RAG). In the context of baseline surveillance, samples are collected through a country-wide network of almost 50 sentinel PCR laboratories (Figure 3), geographically dispersed across Belgium and selected to be representative of the heterogeneous population of persons being infected with SARS-CoV-2 (including hospital and private laboratories as well as federal testing platforms). The selection of these participating laboratories ensured a wide community-based sequencing coverage of positive cases in all Belgian provinces (Figure 2), while in the context of active surveillance, samples could be sent to the sequencing laboratories from all diagnostic COVID-19 testing laboratories across Belgium, including testing laboratories of the main airports in Belgium (located in Zaventem and Charleroi) to monitor import events. Apart from returning travelers, active surveillance was initially focused on reinfection cases, post-vaccination breakthrough samples, atypical PCR results (e.g., dropout or shift in viral load for one of the target genes of a PCR assay), outbreak settings, and populations at risk for mutations (e.g., immunocompromised patients [29]). Throughout the pandemic, there was a specific request to obtain and sequence samples in large outbreak settings, such as elderly care facilities (e.g., [25]), hospitals, or schools, to identify potential superspreaders and fine-tune biosecurity measures. Evolving towards a higher vaccination coverage (see Figure 6), post-vaccination breakthrough cases were asked to be sequenced only in particular situations, such as residents of long-term care facilities or when associated with increased disease severity. Along the road, it was requested that sequencing efforts be reinforced for the hospitalized population, mapping in detail the circulation of variants within hospitals and particular units in detail, allowing for association studies between different VOCs and disease severity ([30,31], but see also Section 4).

As among the national sequencing network, various sequencing protocols and bioinformatics pipelines were implemented to detect and monitor the circulation of SARS-CoV-2 variants, and no official or commercial external quality control assessment was in place for SARS-CoV-2 WGS at the start of the genomic surveillance consortium, quality assurance was decided to be the responsibility of the coordinator of the consortium, i.e., the NRC UZ/KU Leuven, supported by Sciensano. Overall, eight laboratory networks implemented a sequencing protocol relying on Illumina sequencing technology, while nine centers obtained SARS-CoV-2 full-length sequences using the Oxford Nanopore Technology. In the first year of the national genomic surveillance initiative, three cross-validation rounds were organized (see also Figure 5 and Figure 6), in which all sequencing laboratories with a signed convention in a context of reimbursement were required to participate. All sequencing centers were asked to supply the NRC with three to five samples of sufficient leftover material after initial sequencing at their facility, corresponding to different SARS-CoV-2 variants. Sequencing information of the initial sequencing center was also shared with the NRC to facilitate the evaluation of the cross-validation round. At the level of the NRC, new panels were constructed and sent out to the participating sequencing laboratories, consisting of a mix of SARS-CoV-2 variants, different transport media, and with each sample being ideally sent to at least two laboratories (that make use of different sequencing technologies) to render a broad comparison of results possible. The participating laboratories were asked to analyze the received panel according to their standard operating procedures and to report the results within a turnaround of seven working days (as agreed upon in the convention) to the NRC. A thorough evaluation of the results occurred at the level of the NRC, and a report with action points (when appropriate) was shared with all the participating laboratories. Apart from quality assurance, monthly meetings were organized with all members of the sequencing consortium by the NRC UZ/KU Leuven to discuss the setup of the surveillance and associated sequencing coverage of positive cases, data reporting, joint projects, and publications, as well as active troubleshooting. To facilitate joint projects and publications within the large group of partners within the consortium, ethical approval was obtained by the NRC UZ/KU Leuven as coordinating center and with all other centers as co-investigators (S66037). To inform a broad audience, and especially policymakers, of the detailed and near real-time circulation of SARS-CoV-2 variants in Belgium, weekly reports were written and published on the website of the Department of Laboratory Medicine, UZ Leuven, starting 17 January 2021, and these are ongoing today (https://www.uzleuven.be/nl/laboratoriumgeneeskunde/genomic-surveillance-sars-cov-2-belgium (accessed on 19 October 2022)). At the time of writing, 102 reports have been written and disseminated.

The sequencing centers that received samples were responsible for reporting the results to the respective laboratories or prescribing clinicians that requested the sequencing analysis, as well as reporting in real-time the variant information to Sciensano, through the national data platform called healthdata.be, using the message ‘LaboratoryTestResultVariants’, through which variant information obtained by both presumptive genotyping (such as the use of marker PCRs) and WGS could be transferred (https://docs.healthdata.be/nl/node/286 (accessed on 8 June 2022)). This dataflow provided for, amongst others, the weekly reporting in the epidemiological bulletin of Sciensano, the real-time reporting on the public dashboard of Sciensano, and reporting to the ECDC. Furthermore, as stated in the convention, the sequencing centers were also responsible for making the actual nucleotide sequences, annotated with minimal metadata (e.g., age, gender, location, and sequencing indication), publicly available through submission to the international database GISAID [15]. These sequences were used to update both global and Belgium-focused Nextstrain builds (e.g., Figure 2) and to transparently inform public health authorities on the circulation of the different variants of the virus (e.g., results of these analyses are often used in the weekly reports of the consortium). The combination of an extensive spatial sequencing coverage and the integration of a sufficiently representative number of positive samples allowed for the minimization of noise inevitably generated by project-centered sequencing initiatives leading to the over-representation of some regions (see Figure 1) or particular patient groups.

## 4. Studies Resulting from the Genomic Surveillance Initiative

As mentioned in the previous section, the genomic sequences generated by the consortium enabled collaboration on many joint publications, which we discuss here in chronological order (of the variants being studied).

Van Goethem et al. [30] compared COVID-19 disease severity between hospitalized patients in Belgium infected with the SARS-CoV-2 variant B.1.1.7 (Alpha) and those infected with previously circulating strains. Employing a causal framework, the authors observed no difference between patients infected with B.1.1.7 and those infected with older strains in terms of disease severity or in-hospital mortality. However, Van Goethem et al. [30] found that the estimated standardized risk of being admitted to an ICU was significantly higher when infected with the B.1.1.7 variant, and that among the younger age group (≤65 years), B.1.1.7 was significantly associated with both severe COVID-19 progression and ICU admission.

Belgian Armed Forces that engaged in missions and operations around the world were systematically screened, pre- and post-mission, for the presence of SARS-CoV-2, including the identification of viral lineages. A study by Pirnay et al. [32] showed that nine distinct viral genotypes were identified in soldiers returning from operations in Niger, Congo, Afghanistan, and Mali. The SARS-CoV-2 lineages identified included the variant of interest (VOI) B.1.525, the variant under monitoring (VUM) A.27, as well as lineages B.1.214, B.1, B.1.1.254, and A. Through contact tracing and phylogenetic analysis, Pirnay et al. [32] showed that the isolation and testing policies implemented by the Belgian military command appear to have been successful in containing the influx and transmission of these distinct SARS-CoV-2 variants into both military and civilian populations. In a follow-up study dedicated to the A.27 SARS-CoV-2 lineage, Kaleta et al. [33] performed Bayesian phylogeographic analyses obtained from national and international databases to reveal an origin of this lineage in Western Africa, and multiple introductions from there initiated a global spread of this lineage. The authors performed neutralization assays to demonstrate an escape of A.27 from convalescent and vaccine-elicited, antibody-mediated immunity, and to show that the therapeutic monoclonal antibody Bamlanivimab, and partially the REGN-COV2 cocktail, fail to block infection by A.27.

At the start of the wave of infections spawned by the emergence of the SARS-CoV-2 Delta VOC (lineage B.1.617.2), Van Elslande et al. [34] found two clusters of Delta infections in a group of 41 Indian nursing students who traveled from New Delhi, India, to Belgium via Paris, France. Upon arrival in Belgium, the students were quarantined in eight different houses. Four houses remained COVID-free during the 24 days of follow-up, while all 27 residents of the other four houses developed an infection during quarantine, including four residents who were fully vaccinated and two residents who were partially vaccinated. Through phylogenetic analysis of the genomic sequences made available through the nationwide genomic surveillance, Van Elslande et al. [34] could confirm that these quarantined house outbreaks were successfully contained and did not lead to secondary community transmission in Belgium.

Cuypers et al. [35] studied the risk factors for fatal COVID-19 post vaccination in three large nursing home outbreaks (20–35% fatal cases) by combining SARS-CoV-2 aerosol monitoring, whole-genome phylogenetic analysis, and immunovirological profiling. Each outbreak was shown to have been likely caused by a single introduction event, though each time with a different variant (Delta, Gamma, and Mu). Employing survival and time-to-event analysis, the authors found four factors as predictors of mortality: age, male sex, SARS-CoV-2 variant, and timing of infection. Further, Cuypers et al. [35] show that dementia or peak viral load were not predictive of fatal cases in the joint analysis of the three outbreaks but were significant predictors in single nursing homes.

As a result of intensifying genomic surveillance efforts in Belgium, Vanmechelen et al. [36] were able to identify the first infection within Europe with SARS-CoV-2 Omicron VOC (lineage B.1.1.529/BA.1), from a Belgian patient with a history of recent travel to Egypt. This first detection of the Omicron VOC further enabled growing an isolate in cell culture to determine its sensitivity to nine monoclonal antibodies, as well as to antibodies present in 115 serum samples from COVID-19 vaccine recipients or individuals who have recovered from COVID-19 [37]. The authors found that Omicron was completely or partially resistant to neutralization by all monoclonal antibodies tested. Further, they showed that sera from recipients of the BNT162b2 (Pfizer) or ChAdOx1 (AstraZeneca) vaccine, sampled five months after complete vaccination, barely inhibited Omicron, and that sera from COVID-19-convalescent patients collected 6 or 12 months after symptoms displayed low or no neutralizing activity against Omicron. Similar low or absent neutralizing antibody activity was observed for BA.1 versus original lineage B (Wuhan-Hu-1) and Delta (B.1.617.2) in sera from individuals infected with SARS-CoV-2 prior to vaccination, from infection-naïve individuals after three doses of BNT162b2 and from previously-infected individuals after three doses of BNT162b2 [38]. Subsequent early detection of the BA.2 lineage allowed Bruel et al. [39] to compare its sensitivity to neutralization by the aforementioned nine therapeutic monoclonal antibodies against that of BA.1. The authors analyzed sera from 29 immunocompromised individuals up to one month after administration of the Ronapreve (casirivimab and imdevimab) and/or Evusheld (cilgavimab and tixagevimab) antibody cocktails. Bruel et al. [39] showed that all treated individuals displayed elevated antibody levels in their sera, which efficiently neutralized the Delta VOC. Sera from Ronapreve recipients did not neutralize lineage BA.1 and weakly inhibited lineage BA.2, whereas neutralization of BA.1 and BA.2 was detected in 19 and 29 out of 29 Evusheld recipients, respectively.

Starting in late 2020, Alpha was the first VOC to become dominant in Belgium, reaching a relative abundance of up to 82%, whereas the Beta and Gamma VOCs only reached a maximum abundance of 7.7% and 9.9%, respectively. The Belgian COVID-19 epidemic was then followed by waves where Delta, BA.1 (and descendants) and BA.2 (and descendants) were dominant. The replacement of Delta with the Omicron VOC as the dominating lineage worldwide (Figure 5) resulted in possible co-infections of patients by different SARS-CoV-2 strains. Wawina-Bokalanga et al. [40] report on such a co-identification of SARS-CoV-2 variants Omicron and Delta in two geographically unrelated cases. Both patients reported no recent travel history abroad. Such co-infection cases with different variants may lead to the emergence of novel SARS-CoV-2 recombinant variants, which might influence viral transmission, disease severity, and vaccine efficacy [41]. Further, Van Goethem et al. [31] assessed the risk for severe COVID-19, ICU admission, and in-hospital mortality in hospitalized patients when infected with the Omicron variant compared to when infected with the Delta variant. Using data from 954 COVID-19 patients, of which 445 were infected with Omicron, the authors employed a causal framework to show that the estimated standardized risk for severe COVID-19 and ICU admission in hospitalized patients was significantly lower when infected with the Omicron variant, whereas in-hospital mortality was not significantly different according to the SARS-CoV-2 variant.

## 5. Conclusions and Discussion

We have here presented an overview through time of genomic surveillance efforts in Belgium, which evolved from non-structural, project-based initiatives to a nationwide approach following the establishment of a genomic surveillance consortium. We have shown that initiating this consortium at the start of 2021 quickly resulted in a marked increase in the number of SARS-CoV-2 genomic sequences being generated in Belgium, as well as overall sequencing coverage of positive cases (Figure 5). We have also provided an overview of the number of tests performed, alongside the number of cases detected, and the number of recorded hospitalizations, which are obviously all strongly correlated (Figure 6). Of note, the Belgian Interministerial Public Health Conference asked the ECDC to carry out an external evaluation of the testing policy applied by Belgium during June–December 2021, in response to the coronavirus disease 2019 (COVID-19) pandemic [42]. As with the genomic surveillance, the ECDC noted that testing strategies were first decided upon in Belgium by existing institutions, but during the crisis, other bodies were set up; testing strategies were developed based on scientific advice and have regularly been adapted, based on the epidemiological situation. More importantly, and related to this review, the ECDC found that Belgium performs comprehensive genomic surveillance and acknowledged that sequences were reported weekly, with volumes sufficient to estimate variant proportions of 2.5% or lower, following ECDC recommendations.

A similar benchmark was used recently in a pre-print to show that on average, genome surveillance programs in high income countries should be able to detect circulating virus lineages at 5% prevalence with maximum probability under the assumption of random sampling, as shown by Brito et al. [28] The authors have shown that when attaining a sequencing percentage of 5% per week—the target imposed upon initializing the nationwide genomic surveillance in Belgium—there is a 48% probability of detecting a viral lineage before it reaches 100 cases randomly selected from the population, provided a turnaround of 7 days can be achieved. However, this probability can drop markedly with sequencing coverage of positive cases. According to Brito et al. [28], when the proportion of sequenced cases per week decreases by 100-fold, to 0.05%, the probability of the timely detection of a viral lineage before it reaches 100 cases decreases to 4.8% for a turnaround of 7 days, and further declines to 2.6% when turnaround time is 35 days.

Related to turnaround times for global data sharing, a retrospective analysis [43] suggested that some of the key variant-defining mutations could potentially have been detected much earlier, which shows the importance of sharing data rapidly and globally, opening up the possibility of analyzing the data with a series of bioinformatics tools. However, genomic data are not the sole source of information that contribute to the development and implementation of a global risk-monitoring framework for SARS-CoV-2 variants, as this requires a multidisciplinary approach that includes in silico, virological, clinical, and epidemiological (meta)data [44].

In conclusion, the genomic surveillance consortium launched at the start of 2021 has put Belgium in a solid position to attain an adequate level of genomic sequencing coverage of positive cases, ensuring a fair probability of detecting circulating virus lineages. Of note, turnaround times for global data sharing were not assessed at this time, given the only recent publication of guidelines of sequencing coverage and turnaround times [28]. While a systematic study of such turnaround times for SARS-CoV-2 genomic sequencing is beyond the scope of this review, the combination of sequencing a sufficiently large number of cases in combination with short turnaround times hence remains very important to monitor for the foreseeable future in this pandemic.

## Figures and Tables

**Figure 1 viruses-14-02301-f001:**
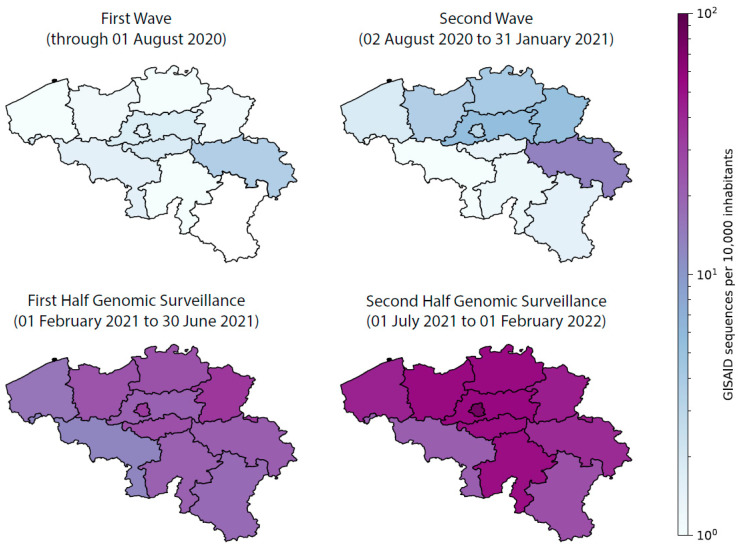
Number of available SARS-CoV-2 genomes for Belgium—per 10,000 inhabitants—on the public GISAID database since the start of the COVID-19 pandemic.

**Figure 2 viruses-14-02301-f002:**
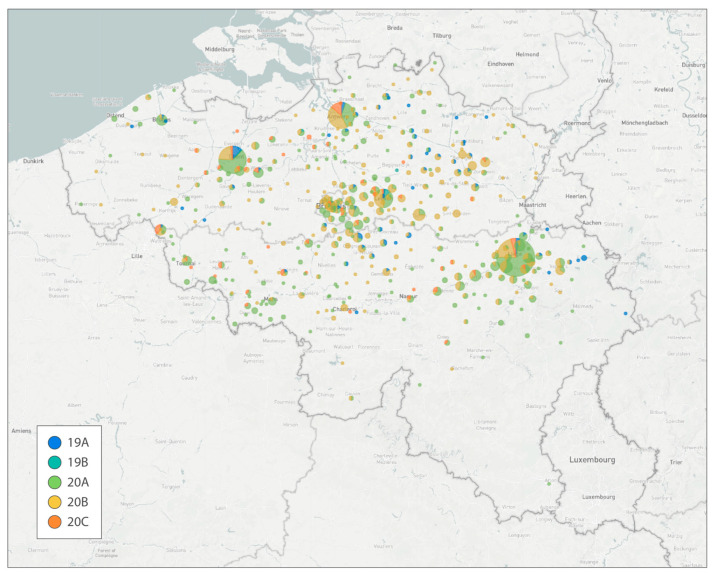
First publicly available Nextstrain build for Belgium, released on 8 January 2021 (just before the Alpha variant was detected in Belgium—or at least before any genomes from Alpha were available). In cases where GDPR prevents sharing the patient’s actual municipality of residence the sequencing center’s location was used, leading to larger numbers for the cities of Leuven, Liège, Ghent, and Antwerp. Pie charts are colored according to the SARS-CoV-2 clades defined in Nextstrain.

**Figure 3 viruses-14-02301-f003:**
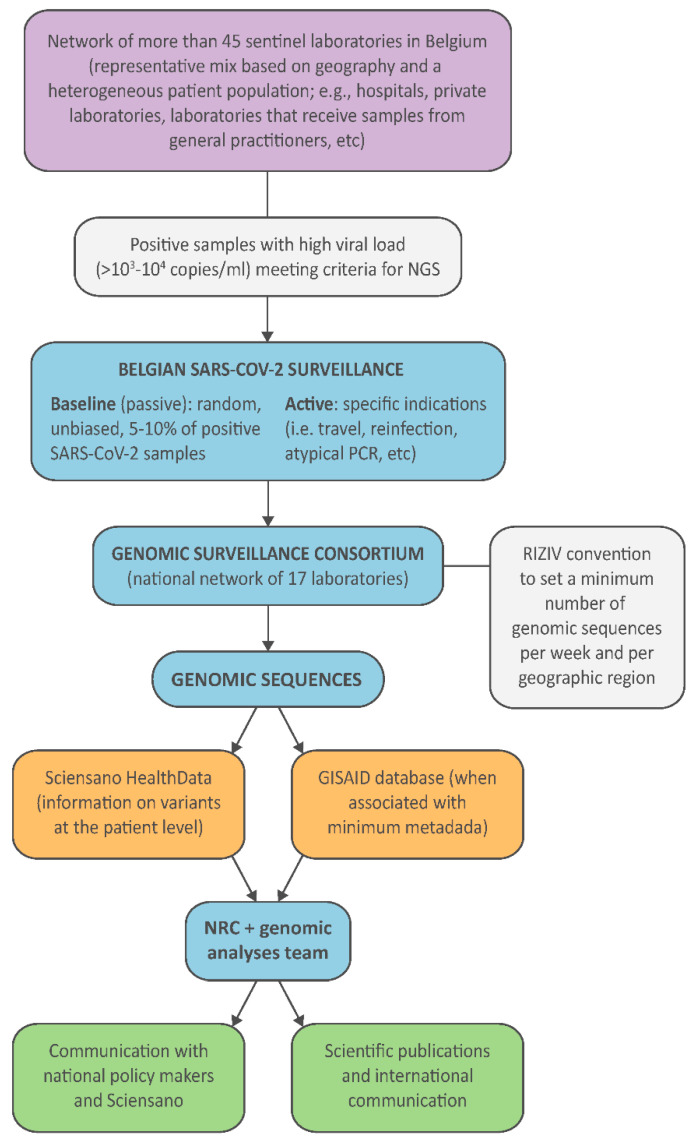
Workflow of the SARS-CoV-2 genomic surveillance established in Belgium. “NRC” stands for National Reference Center (and in this context refers to the NRC for respiratory pathogens), “RIZIV” for National Institute for Health and Disability, and “NGS” for next-generation sequencing.

**Figure 4 viruses-14-02301-f004:**
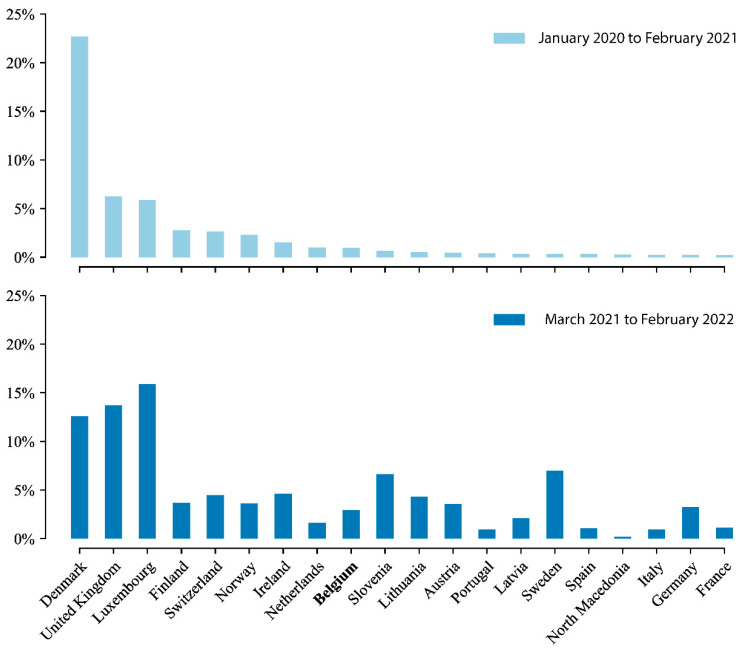
Sequencing efforts for European countries that reported over 10,000 positive cases on a given year, expressed as the percentage of confirmed COVID-19 cases sequenced that have been submitted to GISAID (retrieved from the GISAID database on 31 August 2022), with a minimum sequence/case ratio of 0.1%. Top: Sequencing coverage of positive cases prior to the establishment of the COVID-19 Genomics Belgian Consortium, from January 2020 to the end of February 2021. Bottom: Sequencing coverage of positive cases a year after the establishment of the consortium, from 1 March 2021 to the end of February 2022. Large discrepancies can be seen among European countries, both before and after March 2021, with Belgium’s performance in 2021 at a similar level as most of its neighboring countries.

**Figure 5 viruses-14-02301-f005:**
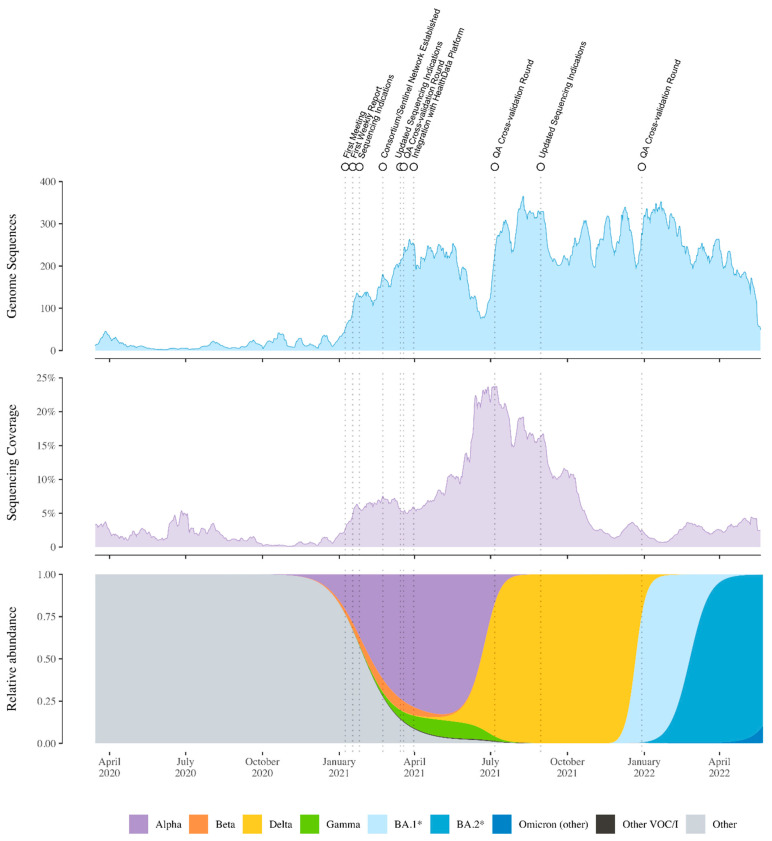
Sequencing numbers, sequencing coverage (the ratio between genomes sequenced and confirmed cases), and relative abundance of SARS-CoV-2 variants of concern in Belgium. Individual colors were assigned to lineages depending on their VOC status and prevalence in Belgium. Dotted lines indicate events related to the COVID-19 Belgium Genomics Consortium.

**Figure 6 viruses-14-02301-f006:**
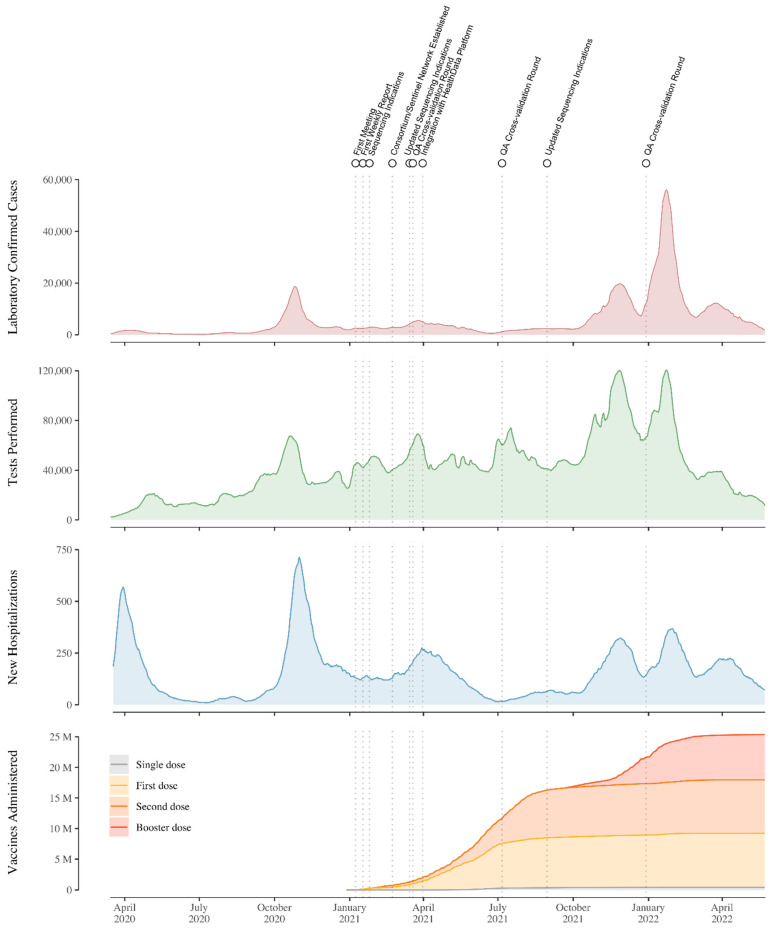
Daily number of confirmed cases (7-day moving average), daily number of COVID-19 tests performed (7-day moving average), new hospitalizations (new daily admissions to hospital of lab-confirmed COVID-19 patients, with a 7-day moving average) and cumulative number of vaccines administered in Belgium from March 2020 to May 2022. Data retrieved from Sciensano, Belgium’s national public health institute (https://epistat.sciensano.be/covid (accessed on 22 August 2022)). Dotted lines indicate events related to the COVID-19 Belgium Genomics Consortium. Peaks in cases and hospitalizations roughly align with moments where the different VOCs were dominant.

## Data Availability

Not applicable.

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
