# Peer review of "Two Years of Genomic Surveillance in Belgium during the SARS-CoV-2 Pandemic to Attain Country-Wide Coverage and Monitor the Introduction and Spread of Emerging Variants"

_viruses, 2022, doi:10.3390/v14102301_

Round 1
Reviewer 1 Report
This review discussed the two-year efforts of SARS-CoV-2 genomic surveillance in Belgium, aiming at reviewing the development and establishment of the surveillance systems, the achievements obtained, as well as studies derived from this nationwide effort. Overall, this review is clearly organized and well written. Please see below my questions and suggestions:
1. I suggest the author to define the word “coverage/sequencing coverage” throughout the text. It is distracting as “sequencing coverage” is a technical term describing sequencing outcomes (i.e., depth). By saying “sequencing coverage”, do the authors mean geographical region coverage, or percent of tested cases? It looks like more for geographical coverage meaning in Figure 1 (line 210), however when it comes to Figure 5 (line 374-375), it seems “sequencing coverage” refers to the percent of tested cases. Similarly, “country-wide coverage” in the title should be re-phrased.
2. Line 276-285: This summary of a patient-specific study can be re-organized to better fit in the flow of the “specific settings”(mentioned in line 266), as the stories next to it are more like epidemiology/transmission studies under specific environments.
3. Line 349-362: I suggest bringing up the efforts of Belgium first, then describe other European countries. This way it will be stronger in the content and more consistent in writing.
4. Line 413: How do the author define the effect(s) of vaccination implementation in Belgium in case numbers and hospitalization rates? Again, please be more detailed for “vaccination coverage” e.g., population.
5. line 463-466: This should be main text, not the figure legend.
6. Figure 6: Please define “New hospitalizations”. Also, dash lines are not defined in the figure. I would also suggest still to include the Figure 5 variant figure in Figure6, or combine Figure 5 and 6, the figure size can be adjusted to fit in to one page.
7. Line 489: Please correct the format of the GISAID reference.
Author Response
REVIEWER 1
This review discussed the two-year efforts of SARS-CoV-2 genomic surveillance in Belgium, aiming at reviewing the development and establishment of the surveillance systems, the achievements obtained, as well as studies derived from this nationwide effort. Overall, this review is clearly organized and well written. Please see below my questions and suggestions:
- I suggest the author to define the word “coverage/sequencing coverage” throughout the text. It is distracting as “sequencing coverage” is a technical term describing sequencing outcomes (i.e., depth). By saying “sequencing coverage”, do the authors mean geographical region coverage, or percent of tested cases? It looks like more for geographical coverage meaning in Figure 1 (line 210), however when it comes to Figure 5 (line 374-375), it seems “sequencing coverage” refers to the percent of tested cases. Similarly, “country-wide coverage” in the title should be re-phrased.
ANSWER: We apologise for the confusion. The text has now been edited to be more explicit. We now talk about "spatial sequencing coverage" or "sequencing coverage of positive cases when relevant".
- Line 276-285: This summary of a patient-specific study can be re-organized to better fit in the flow of the “specific settings” (mentioned in line 266), as the stories next to it are more like epidemiology/transmission studies under specific environments.
ANSWER: Following the remark of the Reviewer, we actually decided to discard this paragraph as it was indeed not enough connected with the rest of the text and, more globally, with the content of the present manuscript.
- Line 349-362: I suggest bringing up the efforts of Belgium first, then describe other European countries. This way it will be stronger in the content and more consistent in writing.
ANSWER: We followed the Reviewer's suggestion and inverted the order of those two paragraphs.
- Line 413: How do the author define the effect(s) of vaccination implementation in Belgium in case numbers and hospitalization rates? Again, please be more detailed for “vaccination coverage” e.g., population.
ANSWER: As it is clearly outside of the scope of the present review dedicated to genomic surveillance, we indeed did not enter into the specificities of the impact of the vaccination campaign on the COVID-19 positive cases or hospitalisations incidence. We however noticed a conceptual mistake associated with Figure 6 and modified it accordingly: instead of reporting the evolution of the number of vaccine doses administrated per day, we now report the cumulative number of first, second, and third (booster) doses administrated through time.
- line 463-466: This should be main text, not the figure legend.
ANSWER: This part of the legend has now been moved to the main text.
- Figure 6: Please define “New hospitalizations”. Also, dash lines are not defined in the figure. I would also suggest still to include the Figure 5 variant figure in Figure6, or combine Figure 5 and 6, the figure size can be adjusted to fit in to one page.
ANSWER: Vertical dashed lines are now re-defined on the top of Figure 6 (as they were on top of Figure 5), but we however prefer to keep those two figures as separated to avoid bringing too much information at the same time. "New hospitaliszations" is now explicitly defined in the figure's legend ("new daily admissions to hospital").
- Line 489: Please correct the format of the GISAID reference.
ANSWER: The reference has been corrected.
Reviewer 2 Report
The authors have presented an interesting review about the spread of SARS-CoV-2 during the pandemic in Belgium. The aim of the work is to show how the country managed the pandemic and the advantages of establishing a nationwide genomic surveillance consortium.
The paper is well written and the overall situation is well illustrated.
Author Response
REVIEWER 2
The authors have presented an interesting review about the spread of SARS-CoV-2 during the pandemic in Belgium. The aim of the work is to show how the country managed the pandemic and the advantages of establishing a nationwide genomic surveillance consortium.
The paper is well written and the overall situation is well illustrated.
ANSWER: Thank you very much for the positive assessment of our work.